# Do beer and wine respond to price and tax changes in Vietnam? Evidence from the Vietnam Household Living Standards Survey

Grieve Chelwa,[1] Pham Ngoc Toan,[2] Nguyen Thi Thu Hien,[3] Le Thi Thu,[4] Pham Thi Hoang Anh,[4] Hana Ross [5]

[1]Graduate School of Business, University of Cape Town, Cape Town, South Africa
[2]Invalids and Social Affairs, Ministry of Labour, Hanoi, Vietnam
[3]Department of Economics, Thuongmai University, Hanoi, Vietnam
[4]Healthbridge Foundation of Canada, Hanoi, Vietnam
[5]SALDRU Research Affiliate, University of Cape Town, Cape Town, South Africa

**Correspondence to**
Dr Grieve Chelwa; grievechelwa@gmail.com

## ABSTRACT

**Objective** To provide the first ever published estimates of the price and expenditure elasticities of demand for beer and wine in Vietnam and thereby contribute to policy initiatives aimed at reducing the excessive consumption of alcohol.

**Methods** We use a linear approximation of the Almost Ideal Demand System and data from the Vietnam Household Living Standards Survey for 2010, 2012 and 2014.

**Results** We find that the demand for beer and wine in Vietnam is price and expenditure inelastic with average price elasticities of −0.283 and −0.317 and average expenditure elasticities of 0.401 and 0.156, respectively. That is, we find that beer and wine consumption decline whenever their respective prices increase and their consumption increases whenever expenditure rises.

**Conclusions** The results of the study lend confidence to calls for increased taxation of alcoholic products on public health grounds in Vietnam.

### Strengths and limitations of this study

► This paper provides the first ever published estimates of the price and expenditure elasticities of demand for beer and wine in Vietnam.
► Often, these estimates are provided for broad alcohol categories but we provide separate estimates for beer and wine.
► The paper uses the method pioneered by Angus Deaton that exploits spatial variation of beer and wine prices in Vietnam.
► A limitation is that we do not provide elasticity estimates for spirits, another important alcohol type consumed in Vietnam.
► A second limitation is that we do not provide cross-price elasticities for beer and wine in our analysis.

## INTRODUCTION

The use of alcohol in Vietnam measured in pure alcohol equivalent per person increased significantly from 3.8 L in 2003–2005 to 6.6 L in 2008–2010.[1] Excessive beer and wine consumption is not only harmful to health, it also increases domestic violence and traffic accidents. Approximately 33.7% of family violence cases in Vietnam are caused by the use of alcohol,[2] while 36.2% and 0.7% of traffic accidents involving men and women were related to alcohol, respectively.[3] About 36% of motorcyclists and 66.8% of car drivers involved in the 18 412 traffic accidents from July 2009 to October 2010 had higher than allowed level of blood alcohol concentration.[4]

To reduce the harmful effects of the excessive use of alcohol, it is important to implement measures to reduce their consumption.[5] Evidence from many countries demonstrates that tax increases resulting in higher prices are one of the most effective measures to control the excessive consumption of alcohol and the associated harms.[6 7] Gallet assessed the size of the price and income effects on alcohol demand by conducting a meta-analysis of 132 international studies and concluded that the price elasticities of beer and wine were about −0.83 and −1.11, respectively, while the income elasticities were about 0.2 and 0.71, respectively.[8] He concluded that the estimated responsiveness to price depends on the choice of demand functions, the model, the estimation method and the type of data used. He pointed out that non-linear demand functions tend to generate smaller elasticities compared with linear demand functions. Bundit et al[9] summarised results of 12 studies estimating the demand for alcohol, including studies conducted in China, India, the Philippines, Taiwan and Thailand. The price elasticities of total alcohol ranged from −0.02 in the Philippines to −1.53 in China.[9]

To assess the impact of tax policy on alcohol consumption in Vietnam, it is important to estimate price and income elasticities of alcohol demand, which indicate the level of

consumers' sensitivity to price and income changes. In this paper we proxy for income using expenditure and therefore estimate expenditure elasticities of demand. Our study focuses on beer and wine, given that together they constitute much of the alcohol consumed in Vietnam. This study is the first to estimate elasticities for beer and wine in Vietnam using the Vietnam Household Living Standards Surveys (VHLSS) of 2010, 2012 and 2014.

## METHOD

The description of the method given here borrows from Chelwa.[10] We use data from the nationally representative VHLSS for 2010, 2012 and 2014. The VHLSS is conducted by the General Statistics Office in Vietnam and informs much of the country's planning processes. Each wave includes data on household income, expenditure and other household characteristics for 9399 households in 3133 clusters (or communes as they are known in Vietnam). A commune is the lowest administrative unit in Vietnam. We focus on the quantity and the monetary value of beer and wine bought in the previous month.

Given that we only have cross-sectional household data without prices and that the same alcohol tax is applied across Vietnam, we apply the Almost Ideal Demand System (AIDS)[11] and a unit value equation[11 12] to estimate elasticities. The main identifying assumption is that prices vary at the cluster level, which in Vietnam is at the commune level. This assumption is reasonable given that households that are geographically proximate, ie, in the same commune, face similar prices in low and middle income countries. This assumption generates exogenous variation in prices which is then used to estimate the elasticities.

Given that the VHLSS does not have price information, we proxy for prices using 'unit values' calculated as:

$$v_{ic} = \frac{y_{ic}}{q_{ic}} \qquad (1)$$

where $v_{ic}$, $y_{ic}$ and $q_{ic}$ are, respectively, the unit value, expenditure and quantity of wine/beer by household $i$ living in commune $c$. However, unit values are not the same thing as prices. They hide quality differences and are subject to measurement error. The method used in this paper (described in full below) addresses these shortcomings in using unit values in place of prices.

The method proceeds in a series of steps. First one conducts an analysis of variance (ANOVA) exercise to divide the total variation in unit values into 'within commune variation' and 'between commune variation'. The ANOVA is conducted to verify our main identifying assumption that prices, unit values in this case, vary at the commune level. A large F-statistic in this exercise leads to the conclusion that unit values vary across communes.

In a second step, one estimates 'within commune' regressions of the form:

$$lnv_{ic} = \lambda + \beta lnx_{ic} + \gamma \mathbf{Z}_{ic} + \psi ln\pi_c + e_{ic} \qquad (2)$$

and

$$w_{ic} = \alpha + \varepsilon lnx_{ic} + \delta \mathbf{Z}_{ic} + \theta ln\pi_c + \left( FE_c + u_{ic} \right) \qquad (3)$$

where $w_{ic}$ is the share of wine/beer expenditure in total household expenditure for household $i$ in commune $c$ and $lnv_{ic}$ is the log of the unit value, derived according to equation (1) for household $i$ in commune $c$. $lnx_{ic}$ is the log of total household monthly expenditure, $\mathbf{Z}_{ic}$ is a vector of household characteristics (household size, gender of household head, age of household head, years of schooling of the household head, employment status of the household head, urban/rural location of the household and ethnicity of the household head), $FE_c$ is a commune fixed effect, and $u_{ic}$ and $e_{ic}$ are the standard regression error terms. We use characteristics of the household head as demographic control variables because household heads are known to influence the household's expenditure decisions. Given that the VHLSS does not contain price information, $ln\pi_c$ are the unobserved prices and consequently, equations (2) and (3) are estimated without them. However, the coefficients on the unobserved price terms can be recovered (see below).

A third step involves stripping the household level demand and unit values of the effects of household expenditure and household characteristics and then averaging across communes. The stripping and averaging is done according to the following equations:

$$\hat{y_c^1} = \frac{1}{n_c} \sum_{i=1}^{n_c} \left( lnv_{ic} - \hat{\beta} lnx_{ic} - \hat{\gamma} \mathbf{Z}_{ic} \right) \qquad (4)$$

and

$$\hat{y_c^2} = \frac{1}{n_c} \sum_{i=1}^{n_c} \left( w_{ic} - \hat{\varepsilon} lnx_{ic} - \hat{\delta} \mathbf{Z}_{ic} \right) \qquad (5)$$

where $n_c$ is number of households in commune $c$, $\hat{y_c^1}$ and $\hat{y_c^2}$ are the estimates of, respectively, commune average unit value and commune average demand after removing the effects of household expenditure and household characteristics.

The next step involves regressing commune level demand, $\hat{y_c^2}$, on commune level unit values, $\hat{y_c^1}$. The coefficient on $\hat{y_c^1}$ in such a regression can be obtained by dividing the covariance between $\hat{y_c^2}$ and $\hat{y_c^1}$ by the variance of $\hat{y_c^1}$:

$$\hat{\phi} = \frac{Cov\left( \hat{y_c^2}, \hat{y_c^1} \right) - \frac{\sigma^{\hat{1}2}}{n_c}}{Var\left( \hat{y_c^1} \right) - \frac{\sigma^{\hat{1}1}}{n_c^+}} \qquad (6)$$

where $n_c^+$ is the number of households in a commune reporting positive expenditures on tobacco and $n_c$ is the number of households in a commune; $\sigma^{\hat{1}2}$ is the estimate of the covariance of the errors in equations (2) and (3); $\sigma^{\hat{1}1}$ is the variance of the errors in equation (2). The variance and covariance of the error terms correct for measurement error in unit values.

The final step applies quality correction formulas in obtaining the estimate of the price elasticity of demand, $\hat{\varepsilon}_P$ :

$$\hat{\varepsilon}_P = \left(\frac{\hat{\theta}}{\bar{w}}\right) - \hat{\psi} \qquad (7)$$

where $\bar{w}$ is the average share of total household expenditure dedicated to wine/beer in the sample, and $\hat{\psi}$ and $\hat{\theta}$ are the estimates of the coefficients on the unobserved price terms in equation (2) and (3), respectively. $\hat{\psi}$ and $\hat{\theta}$ are recovered as follows:

$$\hat{\psi} = 1 - \frac{\hat{\beta}\left(\bar{w} - \hat{\theta}\right)}{\hat{\varepsilon} + \bar{w}} \qquad (8)$$

and

$$\hat{\theta} = \frac{\hat{\phi}}{1 + \left(\bar{w} - \hat{\phi}\right)\hat{\zeta}} \qquad (9)$$

with

$$\hat{\zeta} = \frac{\hat{\beta}}{\hat{\varepsilon} + \bar{w}\left(1 - \hat{\beta}\right)} \qquad (10)$$

where $\hat{\beta}$ is the estimate of the coefficient on total household expenditure in equation (2), the within commune unit value equation, and $\hat{\varepsilon}$ is the coefficient on total household expenditure in equation (3), the within commune demand equation.

The expenditure elasticities of demand $\hat{\varepsilon}_e$ is calculated according to Deaton[12]:

$$\hat{\varepsilon}_e = 1 + \left(\frac{\hat{\varepsilon}}{\bar{w}}\right) - \hat{\beta}. \qquad (11)$$

We estimate elasticities separately for each VHLSS wave as well for the pooled sample. The SEs for the elasticity estimates are obtained by the method of the bootstrap.

Table 1 reports some summary statistics for the variables used in the study. The average household budget shares allocated to wine and beer were respectively 4.5% and 6.5% over the period under study. The reported average unit values were VND17400 (approximately US$0.80) for wine and VND19700 (approximately US$0.92) for beer. VND stands for Vietnamese Dong, Vietnam's currency.

**Table 1** Summary Statistics from the Vietnam Household Living Standards Surveys

| | Mean | SD | Mean | SD | Mean | SD | Mean | SD |
|---|---|---|---|---|---|---|---|---|
| | 2010 | | 2012 | | 2014 | | 2010–2014 | |
| Average age of Household Head | 48.3 | 14.2 | 49.7 | 14.2 | 50.7 | 14.1 | 49.6 | 14.2 |
| Gender of household head (1: male, 0: female) | 0.8 | 0.4 | 0.8 | 0.4 | 0.7 | 0.4 | 0.7 | 0.4 |
| Average household size | 3.9 | 1.6 | 3.9 | 1.6 | 3.8 | 1.6 | 3.9 | 1.6 |
| Ethnicity of household head (1: ethnic minority; 0: otherwise) | 0.17 | 0.38 | 0.17 | 0.38 | 0.17 | 0.37 | 0.17 | 0.37 |
| Average year of schooling of household head | 7.58 | 4.40 | 7.60 | 4.40 | 7.74 | 4.41 | 7.64 | 4.41 |
| Location of household (1: urban, 0: rural) | 0.28 | 0.45 | 0.29 | 0.45 | 0.30 | 0.46 | 0.29 | 0.45 |
| Household head in wage employment (1: yes; 0: no) | 0.41 | 0.49 | 0.39 | 0.49 | 0.40 | 0.49 | 0.40 | 0.49 |
| Average wine expenditure (in thousands of VND) | 38.4 | 56.4 | 46.7 | 61.5 | 51.1 | 48.8 | 45.2 | 56.3 |
| Average beer expenditure (in thousands of VND) | 98.7 | 111.9 | 117.4 | 108.6 | 140.5 | 133.9 | 118.1 | 119.6 |
| Average unit value for wine (in thousands of VND) | 14.2 | 31.1 | 18.2 | 34.7 | 20.1 | 15.2 | 17.4 | 28.8 |
| Average unit value for beer (in thousands of VND) | 16.1 | 10.2 | 19.9 | 9.7 | 23.6 | 12.0 | 19.7 | 11.1 |
| Average household expenditure (in thousands of VND) | 1054.9 | 769.7 | 1447.7 | 1002.3 | 1508.6 | 1037.6 | 1337.0 | 965.2 |
| Share of total household expenditure dedicated to wine | 0.049 | 0.062 | 0.042 | 0.048 | 0.044 | 0.053 | 0.045 | 0.055 |
| Share of total household expenditure dedicated to beer | 0.068 | 0.062 | 0.060 | 0.057 | 0.067 | 0.056 | 0.065 | 0.058 |

Source, calculated from VHLSS.
VHLSS, Vietnam Household Living Standards Surveys; VND, Vietnamese Dong.

**Table 2** Wine and beer unit value regressions

| | Wine | | | | Beer | | | |
|---|---|---|---|---|---|---|---|---|
| **Variables** | **2010** | **2012** | **2014** | **Pooled** | **2010** | **2012** | **2014** | **Pooled** |
| Lnx | 0.060*** (0.009) | 0.070*** (0.010) | 0.066*** (0.011) | 0.137*** (0.006) | 0.316*** (0.021) | 0.238*** (0.022) | 0.213*** (0.021) | 0.343*** (0.012) |
| Gender | −0.025 (0.016) | −0.018 (0.016) | 0.005 (0.018) | −0.003 (0.010) | −0.015 (0.027) | −0.001 (0.027) | 0.011 (0.026) | 0.008 (0.016) |
| Lnage | 0.030 (0.021) | 0.131*** (0.023) | 0.082*** (0.026) | 0.118*** (0.014) | −0.066 (0.042) | −0.069 (0.045) | −0.095** (0.045) | −0.048* (0.026) |
| Lnschool | 0.111*** (0.009) | 0.155*** (0.010) | 0.164*** (0.011) | 0.133*** (0.006) | −0.003 (0.023) | −0.008 (0.022) | −0.006 (0.020) | −0.027** (0.013) |
| Employment | 0.005 (0.012) | 0.025* (0.013) | −0.002 (0.013) | 0.015* (0.008) | −0.027 (0.025) | −0.014 (0.026) | −0.044* (0.025) | −0.019 (0.015) |
| Lnsize | −0.030** (0.015) | −0.064*** (0.016) | −0.025 (0.017) | −0.093*** (0.010) | −0.192*** (0.032) | −0.183*** (0.031) | −0.142*** (0.031) | −0.224*** (0.019) |
| Urban | 0.117*** (0.013) | 0.109*** (0.014) | 0.137*** (0.015) | 0.096*** (0.009) | 0.053** (0.024) | 0.115*** (0.023) | 0.128*** (0.022) | 0.074*** (0.014) |
| Ethnic | −0.028* (0.015) | −0.023 (0.015) | −0.053*** (0.017) | 0.011 (0.010) | 0.076* (0.040) | 0.050 (0.044) | 0.029 (0.048) | 0.055** (0.026) |
| Constant | 1.851*** (0.102) | 1.564*** (0.114) | 1.808*** (0.129) | 1.170*** (0.069) | 0.909*** (0.215) | 1.586*** (0.236) | 1.971*** (0.242) | 0.836*** (0.134) |
| Observations | 4241 | 4199 | 3758 | 12 198 | 1918 | 1717 | 1705 | 5340 |
| R² | 0.112 | 0.138 | 0.151 | 0.149 | 0.156 | 0.120 | 0.109 | 0.179 |

***P<0.01; **P<0.05; *P<0.1.
Robust SEs in parenthesis.
Employment, employment status of household head; Ethnic, ethnic group of household head; Gender, gender of household head; Lnage, natural logarithm of age of household head; Lnschool, natural logarithm of years of schooling of household head; Lnsize, natural logarithm of household size; Lnx, natural logarithm of household expenditure; Urban, location of household.

## Patients and public involvement

Patients and the public were not involved in this study.

## RESULTS

The results of the ANOVA exercise are discussed here but not reported for reasons of space. They are, however, available from the authors upon request. The R squared from the ANOVA exercise indicates that at least 85% of the variation in unit values for wine and beer is attributable to variations between communes. The F statistics and the associated p values allow us to reject the null hypothesis of no spatial variation in 2010, 2012, 2014 and pooled data. In other words, prices vary significantly between communes in Vietnam.

The results of the unit value regressions (equation (2)) are reported in table 2. The results show that households with higher expenditure report higher unit values. In other words, unit values reflect choices about quality. These so-called expenditure elasticities of quality range between 0.06 and 0.07 for wine and between 0.22 and 0.32 for beer. This means that the reported unit values of wine and beer rise by 0.6%–0.7% and by 2.2%–3.2% for every 10% increase in household expenditure, respectively.

Households with older household heads and with heads with more schooling report higher unit values for wine.

The impact of household size is negative for both wine and beer, implying that larger households report lower unit values. Households living in urban areas tend to have higher unit values for both alcohol products while ethnic minorities report lower unit values for wine.

Results of regressing wine and beer budget shares on household expenditure and other household characteristics (equation (3)) are reported in table 3. The results show that there is a negative and statistically significant relationship between household expenditure and the share of the household budget allocated to wine and beer. More years of schooling among household heads is related to a smaller budget share on both wine and beer. Households with a male head, located in urban areas and from an ethnic minority devote larger budget shares to wine.

The price elasticity estimates, $\hat{\varepsilon}_P$, for 2010, 2012, 2014 and the pooled sample are presented in table 4.

All estimates are statistically significant at the 1% level except for wine in 2014. The price elasticity of demand for wine ranges from −0.237 to −0.491 with the pooled estimate centring on −0.317. This means that the demand for wine in Vietnam is expected to decline on average by 3.17% for every 10% rise in wine prices. The price elasticity of demand for beer ranges from −0.251 to −0.346

**Table 3** Wine and beer budget share regressions

| | Wine | | | | Beer | | | |
|---|---|---|---|---|---|---|---|---|
| **Variables** | **2010** | **2012** | **2014** | **Pooled** | **2010** | **2012** | **2014** | **Pooled** |
| Lnx | −0.037*** (0.004) | −0.027*** (0.002) | −0.038*** (0.004) | −0.032*** (0.002) | −0.013*** (0.005) | −0.023*** (0.007) | −0.018*** (0.004) | −0.017*** (0.003) |
| Gender | 0.002 (0.003) | 0.004** (0.002) | 0.006*** (0.002) | 0.004*** (0.001) | 0.004 (0.004) | 0.003 (0.003) | 0.002 (0.003) | 0.003* (0.002) |
| Lnage | −0.009*** (0.003) | −0.002 (0.003) | 0.001 (0.004) | −0.003 (0.002) | −0.011 (0.007) | −0.006 (0.006) | −0.007 (0.006) | −0.007* (0.004) |
| Lnschool | −0.001 (0.001) | −0.005*** (0.001) | −0.001 (0.001) | −0.002*** (0.001) | −0.005 (0.003) | −0.013*** (0.003) | −0.012*** (0.003) | −0.011*** (0.002) |
| Employment | −0.009*** (0.002) | −0.003* (0.001) | −0.006*** (0.002) | −0.006*** (0.001) | −0.002 (0.004) | −0.000 (0.003) | −0.000 (0.003) | −0.000 (0.002) |
| Urban | 0.004** (0.002) | 0.005*** (0.002) | 0.007*** (0.002) | 0.004*** (0.001) | 0.001 (0.004) | 0.008 (0.005) | −0.003 (0.003) | 0.002 (0.003) |
| Ethnic | 0.018*** (0.002) | 0.019*** (0.002) | 0.018*** (0.002) | 0.019*** (0.002) | 0.006 (0.006) | 0.009 (0.006) | −0.009 (0.006) | 0.003 (0.004) |
| Constant | 0.326*** (0.029) | 0.244*** (0.019) | 0.292*** (0.028) | 0.274*** (0.015) | 0.240*** (0.048) | 0.296*** (0.051) | 0.281*** (0.038) | 0.261*** (0.027) |
| Observations | 4241 | 4199 | 3758 | 12 198 | 1918 | 1717 | 1705 | 5340 |
| $R^2$ | 0.253 | 0.236 | 0.270 | 0.247 | 0.047 | 0.086 | 0.086 | 0.066 |

***P<0.01 ; **P<0.05 ; *P<0.1.

Employment, employment status of household head; Ethnic, ethnic group of household head; Gender, gender of household head; Lnage, natural logarithm of age of household head; Lnschool, natural logarithm of years of schooling of household head; Lnsize, natural logarithm of household size; Lnx, natural logarithm of household expenditure; Urban, location of household.

with the pooled estimate centring on −0.283. This means that the demand for beer is expected to decline by 2.83% on average for every 10% rise in beer prices.

The estimates of the expenditure elasticities, $\hat{\varepsilon}_e$, for 2010, 2012 and 2014 are reported in table 5. As in table 4, all estimates are statistically significant at the 1% level except for wine in 2014. For the pooled data, the expenditure elasticities are 0.156 and 0.401 for wine and beer, respectively. A 10% increase in expenditure leads to a 1.6% and 4.0% increase, on average, in the demand for wine and beer, respectively. This shows that increases in

household income, proxied here by household expenditure, will increase wine and beer consumption in Vietnam.

## DISCUSSION

This paper estimates the first ever price and expenditure elasticities of demand for wine and beer in Vietnam using the VHLSS. The statistical analysis revealed that wine and beer in Vietnam are price inelastic. In other words, the quantity demanded of wine and beer responds less than

**Table 4** Estimates of price elasticity of demand for wine and beer

| | Observed coefficient | Bootstrap SE | Z | P>Z | Normal-based (95% CI) | |
|---|---|---|---|---|---|---|
| **Wine** | | | | | | |
| 2010 | −0.237 | 0.085 | −2.800 | 0.005 | −0.403 | −0.071 |
| 2012 | −0.491 | 0.029 | −16.950 | 0.000 | −0.548 | −0.434 |
| 2014 | −0.359 | 1.409 | −0.260 | 0.799 | −3.122 | 2.403 |
| Pooled | −0.317 | 0.030 | −10.560 | 0.000 | −0.376 | −0.258 |
| **Beer** | | | | | | |
| 2010 | −0.251 | 0.025 | −10.140 | 0.000 | −0.300 | −0.203 |
| 2012 | −0.305 | 0.032 | −9.500 | 0.000 | −0.368 | −0.242 |
| 2014 | −0.346 | 0.029 | −11.850 | 0.000 | −0.404 | −0.289 |
| Pooled | −0.283 | 0.014 | −20.120 | 0.000 | −0.311 | −0.256 |

Source, calculated from VHLSS

VHLSS, Vietnam Household Living Standards Survey.

**Table 5**  Estimates of the expenditure elasticity of demand

| | Observed coefficient | Bootstrap SE | z | P>Z | Normal-based (95% CI) | |
|---|---|---|---|---|---|---|
| **Wine** | | | | | | |
| 2010 | 0.122 | 0.065 | 1.890 | 0.059 | −0.005 | 0.249 |
| 2012 | 0.339 | 0.040 | 8.550 | 0.000 | 0.261 | 0.416 |
| 2014 | 0.097 | 0.080 | 1.210 | 0.225 | −0.060 | 0.255 |
| Pooled | 0.156 | 0.034 | 4.630 | 0.000 | 0.090 | 0.222 |
| **Beer** | | | | | | |
| 2010 | 0.484 | 0.060 | 8.000 | 0.000 | 0.365 | 0.602 |
| 2012 | 0.406 | 0.101 | 4.000 | 0.000 | 0.207 | 0.604 |
| 2014 | 0.503 | 0.053 | 9.420 | 0.000 | 0.398 | 0.607 |
| Pooled | 0.401 | 0.036 | 11.110 | 0.000 | 0.330 | 0.472 |

Source, calculated from VHLSS.
VHLSS,  Vietnam Household Living Standards Survey.

proportionately to price increases. These results are in line with elasticity estimates from other parts of the world.[8]

Our results on the price inelastic nature of the demand for wine and beer has two immediate implications for the authorities in Vietnam. First, it means that tax measures can be used as an effective tool to control the excessive consumption of alcohol in Vietnam. Second, the government can raise additional tax revenue by increasing the level of alcohol taxes in the country.

The results on the expenditure elasticities of demand also have implications for the country. Our estimated elasticities of demand, which are all positive, mean that the quantity demanded of wine and beer in Vietnam increases every time household expenditure (or household income) increases. This means that alcohol consumption will continue to increase, and increase at a fast pace, given that Vietnam is a fast growing economy. This gives further impetus to the use of tax measures in curbing excessive demand especially in an environment of fast rising incomes.

Our paper has three limitations, however. First, we are unable to provide elasticity estimates for spirits, another important alcohol category. This is because we do not have good data on spirits to enable us estimate their elasticities. Second, we are unable to provide cross-price elasticities between wine and beer. Cross price elasticities are important because policymakers often want to know how the demand for one good responds to levying a tax on a close substitute. Lastly, our elasticity estimates are conditional elasticity estimates and not total elasticity estimates. In other words, we only provide demand responses for current drinkers of wine and beer and not for the entire population. In other words, we do not estimate total price elasticities. However, conditional elasticities are often smaller in absolute value than total elasticities. This means that the total demand response to prices is likely to be much higher than the estimates presented in this paper.

**Contributors**  PNT did the analysis with the help of GC. NTTH, HR, LTT and PTHA helped with the writing of the manuscript.

**Funding**  This work was supported by the International Development Research Centre grant number 107200-002.

**Competing interests**  None declared.

**Patient consent for publication**  Not required.

**Ethics approval**  This study did not require ethics approval as it uses the Vietnam Household Living Standards Survey which is a secondary dataset that is publically available.

**Provenance and peer review**  Not commissioned; externally peer reviewed.

**Data sharing statement**  The data contained in the article are publicly available from the General Statistics Office of Vietnam.

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
