## [Reviewer comments · BMJ Open]

ARTICLE DETAILS

TITLE (PROVISIONAL)	Do Beer and Wine Respond to Price and Tax Changes in Vietnam? Evidence from the Vietnam Household Living Standards Survey
AUTHORS	Chelwa, Grieve; Toan, Pham Ngoc; Hien, Nguyen Thi Thu; Thu, Le Thi; Anh, Pham Thi Hoang; Ross, Hana

VERSION 1 - REVIEW

REVIEWER	Heng Jiang La Trobe University
REVIEW RETURNED	06-Nov-2018

GENERAL COMMENTS	This is an interesting paper. Authors estimated price elasticity of alcohol demand and expenditure elasticity of alcohol demand in Vietnam using three waves of household living standards survey data between 2010 and 2014. The study addresses an important research question and have potential to contribute to the future alcohol price/tax policy. There are some concerns in the study, which need to be addressed. Major issues: 1. Survey data, author indicated that each wave includes various data for 9,399 households. It's the same household or not? Whether the data were national representative? What were the response rates of three surveys? Whether this is missing data in the analysis? For income and price change over time, did authors control the inflation or CPI in the analysis before pooled them together?2. It's not clear how the alcohol unit or alcohol demand was defined in the study. Is it quantity of purchased pure alcohol/standard drinks or quantity of alcohol? Percentage of alcohol content in beer and wine could be significantly different.3. I am particularly confused by the sex and age of household. Generally, the household survey asked expenditure, income and unit of consumption based on the all people in the household as a whole. However, age and sex data could be the person who undertook the interview or may be the representative person in the household. It's not been specified in the current paper. It would be good to add a table to show descriptive statistics of the household representatives or interviewees. If the characteristics of household representative person are similar across three waves, then, you could be able to pooled them together.4. What's the cluster and how did you define a cluster?5. It would be good to add some background information about beer and wine consumption and price in the introduction. I would assume Vietnam is mainly a beer drinking country. However,
--

	results in Table 1, household who had beer expenditure is 20% of the total sample and household who had wine expenditure is about 50% of the total sample? 6. The authors aimed to estimate income elasticity of demand. But I couldn't find any information about household income in the paper. Instead, authors estimated expenditure elasticity of demand and I think this is different with what you aimed for. 7. Authors discussed effects of household budget size, education level and variations between urban and rural. However, I could find the related results in the texts or in Tables. 8. More sophisticated discussions on the results are needed. Including study limitations, substitution relationship between alcohol and other goods and services and across different alcoholic beverages. The interpretation of results also need to be cautious, as the elasticities were below 1, meaning the price elasticity of beer and wine demand is elastic. 9. It might be worth to mention some lag effects of price and income change on the alcohol consumption, although it would be unable to measure in this study. Some households had no alcohol expenditure may be due to health, religious, availability or price/affordability reasons and this need to be included in the discussion. Minor issues: Typos in the paper, e.g., abstract-page 2, line 26 "credence". Some English proof reading and improvement are needed. Provide full terms of Abbreviations at the first occurrence in text and Tables, ANOVA? VND?
--	---

REVIEWER	Kimmo Herttua University of Southern Denmark, Denmark
REVIEW RETURNED	08-Nov-2018

GENERAL COMMENTS	The manuscript examines price and income elasticities of demand for beer and wine in Vietnam. The authors use a linear approximation of Almost Ideal Demand System (AIDS) and data from the Vietnam Household Living Standards Survey for 2010, 2012 and 2014. The authors found that these associations are inelastic which means that beer and wine consumption decline whenever their respective prices increase and their consumption increases whenever incomes rise. The article concludes that results of the study give credence to calls for increased taxation of alcohol products on public health grounds particularly in an environment where incomes have been growing like they have in Vietnam. Abstract: Conclusions need to reflect findings and not to give a simple answer to a very complicated policy question. Page 4, the 3rd paragraph. Use of alcohol is not necessary harmful. Most people worldwide use alcohol without harmful effects. It is also well-known that light-to-moderate use of alcohol has many positive effects on health and other outcomes. It is therefore essential to separate misuse or excessive use from normal use of alcohol. This should be acknowledged in this paragraph and this applies to the manuscript also more generally. P. 5. What are clusters? What is VND? Table 1. Sex of head? What does this mean? Explain, please. This table and other tables as well need to be clarified also in other respects.
---

	Pp. 7-9. I am not sure whether this kind of detailed description of methods with so many equations is in line with the policy of the journal. Pp. 7-9. A description of household and other covariates is lacking. Pp. 7-9. A description of statistical analysis is lacking. Pp. 9-12. The Results section is incoherent and needs restructuring. Discussion section is modest. There is, for instance, no discussion about strengths and limitations. The quality of some references is questionable (e.g. reference 2) and the number of references is not sufficient.
--	---

VERSION 1 – AUTHOR RESPONSE

Reviewer(s)' Comments to Author:

Reviewer: 1

Reviewer Name: Heng Jiang

Institution and Country: La Trobe University

Please state any competing interests or state 'None declared': None

Please leave your comments for the authors below

This is an interesting paper. Authors estimated price elasticity of alcohol demand and expenditure elasticity of alcohol demand in Vietnam using three waves of household living standards survey data between 2010 and 2014. The study addresses an important research question and have potential to contribute to the future alcohol price/tax policy. There are some concerns in the study, which need to be addressed.

Major issues:

1. Survey data, author indicated that each wave includes various data for 9,399 households. It's the same household or not? Whether the data were national representative? What were the response rates of three surveys? Whether this is missing data in the analysis? For income and price change over time, did authors control the inflation or CPI in the analysis before pooled them together?

RESPONSE: The three waves are cross-sectional surveys implying that not the same households are surveyed. In other words, the Vietnam Household Living Standards Survey is not a panel survey. Further, the survey is nationally representative and is conducted by the General Statistics Office in Vietnam. The response rates for the survey tend to be high as is the case with large household surveys. Like any survey, the Vietnam Household Living Standards Survey tends to have missing data but fortunately for us the pattern of missing data is not systematic and therefore does not pose a problem for our analysis. And our analysis did control for inflation before pooling datasets together.

2. It's not clear how the alcohol unit or alcohol demand was defined in the study. Is it quantity of purchased pure alcohol/standard drinks or quantity of alcohol? Percentage of alcohol content in beer and wine could be significantly different.

RESPONSE: Beer and wine in this study are defined in terms of standard drinks. This is how they are captured as per the questionnaire in the Vietnam Household Living Standards Survey. And the

reviewer is correct that the percentage of alcohol often differs between beer and wine. It is for this reason that in this study we conducted the analysis separately for beer and wine.

3. I am particularly confused by the sex and age of household. Generally, the household survey asked expenditure, income and unit of consumption based on the all people in the household as a whole. However, age and sex data could be the person who undertook the interview or may be the representative person in the household. It's not been specified in the current paper. It would be good to add a table to show descriptive statistics of the household representatives or interviewees. If the characteristics of household representative person are similar across three waves, then, you could be able to pooled them together.

RESPONSE: All variables in this study, and indeed in the Vietnam Household Living Standards Survey (VHLSS), are captured at the household level. Often the survey interviews a single member of the household who then provides responses for the household as a whole. So the age and sex data in the survey represent age and sex (gender) characteristics of the household as a whole (age of the household head, sex/gender of the household head, proportion of females in household, proportion of males in household, etc...). This is standard practice in how data is collected in household surveys. If you look at Table 1, you will notice that the demographic variables presented there are largely similar across waves. Even if they were not similar, we could still conduct the analysis for as long as we can control for them in the regression analysis, something that we do in our study.

4. What's the cluster and how did you define a cluster?

RESPONSE: A cluster is at the level of a commune in Vietnam. A commune is the lowest administrative unit in the country – it is the so-called third tier administrative unit. Thanks to this comment by the reviewer, we have made the paper much more clearer by continuously reminding the reader that a cluster is a commune in Vietnam.

5. It would be good to add some background information about beer and wine consumption and price in the introduction. I would assume Vietnam is mainly a beer drinking country. However, results in Table 1, household who had beer expenditure is 20% of the total sample and household who had wine expenditure is about 50% of the total sample?

RESPONSE: The reviewer is correct that much of the alcohol consumed in Vietnam is beer. On a per capita basis, for example, beer consumption is higher than wine consumption. However, the results presented in Table 1 report a different statistic. They do not report the quantities consumed, where beer would certainly be higher, but rather report the percentage of households with positive expenditures, however small, on either beer or wine. Prompted by the reviewer's comments and noting that reporting it this way might be confusing for readers, we have decided to remove this particular statistic from Table 1. Second, we have added the following sentence in the introduction:

"In this study we focus on beer and wine, given that together they constitute much of the alcohol consumed in Vietnam."

6. The authors aimed to estimate income elasticity of demand. But I couldn't find any information about household income in the paper. Instead, authors estimated expenditure elasticity of demand and I think this is different with what you aimed for.

RESPONSE: The reviewer is correct in noting this. However, the standard practice in analyses that use household expenditure surveys, such as ours, is to use household expenditure as a proxy for household income given the well-known shortcomings of household income. For example, household income tends to grossly under predict household expenditure given the many different ways households supplement income to have higher expenditure. However, it turns out that elasticities obtained using household expenditure are often equal in magnitude to those obtained using household income. This is a result of the fact that household income is highly correlated with

household expenditure (technically, the variation in household income is similar to the variation in household expenditure). In light of the reviewers very important comment here and to prevent confusing the reader, we have decided to refer to income elasticities as expenditure elasticities throughout the manuscript. However, we still remind the reader that expenditure elasticities can be interpreted as income elasticities given the explanations advanced here.

7. Authors discussed effects of household budget size, education level and variations between urban and rural. However, I could find the related results in the texts or in Tables.

RESPONSE. The reviewer is correct in noting this. Initially we had decided to only discuss the results without reporting the accompanying regression tables. We did this to save on space. However, prompted by this comment we have since included Tables 2 and 3 pertaining to exactly the reviewer's comments.

8. More sophisticated discussions on the results are needed. Including study limitations, substitution relationship between alcohol and other goods and services and across different alcoholic beverages. The interpretation of results also need to be cautious, as the elasticities were below 1, meaning the price elasticity of beer and wine demand is elastic.

RESPONSE: We have rewritten the discussion section to capture some of the concerns raised by the reviewer here. However, on the interpretation of the elasticities, we would like to remind the reviewer that elasticities below 1 (in absolute size) do not mean that demand is elastic. They actually mean the opposite which is that demand is inelastic. This is precisely the interpretation given in this paper.

9. It might be worth to mention some lag effects of price and income change on the alcohol consumption, although it would be unable to measure in this study. Some households had no alcohol expenditure may be due to health, religious, availability or price/affordability reasons and this need to be included in the discussion.

RESPONSE: We thank the reviewer for this comment. However, as the reviewer themselves note, many of the matters the reviewer raises here are empirical questions which our study does not empirically interrogate. We, therefore, cannot comment on them with some level of confidence. We leave it to other researchers to carefully study them, at least within the context of Vietnam.

Minor issues:

Typos in the paper, e.g., abstract-page 2, line 26 "credence".

RESPONSE: The word "credence" is not a typo. It is an English word that means exactly the same thing as "confidence". To prevent confusion, we have replaced the word credence with confidence in the abstract.

Some English proof reading and improvement are needed.

REPNSE: This has been done. One of the co-authors is an English first speaker and has read the entire manuscript and improved it greatly.

Provide full terms of Abbreviations at the first occurrence in text and Tables, ANOVA? VND?

RESPONSE: This has been done. The full terms of every abbreviation are provided the first time the abbreviation is used.

Reviewer: 2

Reviewer Name: Kimmo Herttua

Institution and Country: University of Southern Denmark, Denmark

Please state any competing interests or state 'None declared': None declared

Please leave your comments for the authors below

The manuscript examines price and income elasticities of demand for beer and wine in Vietnam. The authors use a linear approximation of Almost Ideal Demand System (AIDS) and data from the Vietnam Household Living Standards Survey for 2010, 2012 and 2014.

The authors found that these associations are inelastic which means that beer and wine consumption decline whenever their respective prices increase and their consumption increases whenever incomes rise. The article concludes that results of the study give credence to calls for increased taxation of alcohol products on public health grounds particularly in an environment where incomes have been growing like they have in Vietnam.

Abstract: Conclusions need to reflect findings and not to give a simple answer to a very complicated policy question.

RESPONSE: We wish the reviewer had been more specific about which part of the abstract they are referring to. Otherwise, the abstract reflects exactly the findings of our paper, namely that beer and wine respond to price changes in Vietnam. A policy implication of this finding is that the authorities in Vietnam can use excise tax methods to control the excessive consumption of the two commodities. However, we have tampered down some of the language in the abstract to partly reflect some of the reviewer's concerns here.

Page 4, the 3rd paragraph. Use of alcohol is not necessary harmful. Most people worldwide use alcohol without harmful effects. It is also well-known that light-to-moderate use of alcohol has many positive effects on health and other outcomes. It is therefore essential to separate misuse or excessive use from normal use of alcohol. This should be acknowledged in this paragraph and this applies to the manuscript also more generally.

RESPONSE: This has been acknowledged. We have rewritten the paragraph to reflect the distinction between normal and excessive alcohol use. We have also tampered down the language in as far as this comment is concerned throughout the paper.

P. 5. What are clusters? What is VND?

RESPONSE: A cluster is at the level of a commune in Vietnam. A commune is the lowest administrative unit in the country – it is the so-called third tier administrative unit. Thanks to this comment by the reviewer, we have made the paper much more clearer by continuously reminding the reader that a cluster is a commune in Vietnam.

VND refers to the abbreviation for the Vietnamese Dong, the country's currency. We have written the following sentence, the first time VND is used, to explain to readers what it stands for:

VND stands for Vietnamese Dong, Vietnam's currency

Table 1. Sex of head? What does this mean? Explain, please. This table and other tables as well need to be clarified also in other respects.

RESPONSE: Sex of head refers to the gender of the head of the household. We have since rewritten "Sex of head" in Table 1 as "Gender of Household Head" for more clarity.

Pp. 7-9. I am not sure whether this kind of detailed description of methods with so many equations is in line with the policy of the journal.

RESPONSE: Unfortunately the method we use in this paper requires a little more detail to explain it. We, however, have made sure that the intuitive aspects of every line is explained in plain English to allow readers to follow what is going on. The reviewer will be interested to note that the description of the method given here is far less technical than would be found in other research papers that have used this method to conduct similar analyses.

Pp. 7-9. A description of household and other covariates is lacking.

RESPONSE: All the covariates of the regression analysis are described in full in the paragraph immediately following equations (2) and (3).

Pp. 7-9. A description of statistical analysis is lacking.

RESPONSE: The statistical analysis is mainly conducted using regression analysis and this has been spelt out clearly throughout the Method section.

Pp. 9-12. The Results section is incoherent and needs restructuring.

RESPONSE: We thank the reviewer for this comment. We have since rewritten the results section in addition to including two more tables. We are now confident that it now reads better and is more coherent.

Discussion section is modest. There is, for instance, no discussion about strengths and limitations.

RESPONSE: The discussion section has been rewritten to reflect the reviewer's concerns including a discussion of limitations.

The quality of some references is questionable (e.g. reference 2) and the number of references is not sufficient.

RESPONSE: We have removed reference number 2. On the other hand, the number of references in this paper are sufficient for the purposes of the study under question. In any case, the limited number of references reveals the paucity of research work in this area in Vietnam.

VERSION 2 – REVIEW

REVIEWER	Heng Jiang La Trobe University, Australia
REVIEW RETURNED	01-Feb-2019

GENERAL COMMENTS	I can see that authors have revised the paper substantially. Most of my concerns have been addressed. However, I have specific concern on the results of sex/gender of household head. I think may be use the household reference person instead of household head could be more accurate. In common household expenditure survey, a representative person is selected to participant in the survey and the sex of this person could not represent the gender
---

	of the household. Because household is generally a group of people (e.g., family, friends, renters, or single person) living in a same dwelling. Alcohol consumption and spending on different gender are quite different and the results of sex of household head here are problematic and I suggest to delete it.
--	--

VERSION 2 – AUTHOR RESPONSE

RESPONSE TO REVIEWERS

Reviewer(s)' Comments to Author:

Reviewer: 1

Reviewer Name: Heng Jiang

Institution and Country: La Trobe University, Australia

Please state any competing interests or state 'None declared': None declared

Please leave your comments for the authors below

I can see that authors have revised the paper substantially. Most of my concerns have been addressed. However, I have specific concern on the results of sex/gender of household head. I think may be use the household reference person instead of household head could be more accurate. In common household expenditure survey, a representative person is selected to participant in the survey and the sex of this person could not represent the gender of the household. Because household is generally a group of people (e.g., family, friends, renters, or single person) living in a same dwelling. Alcohol consumption and spending on different gender are quite different and the results of sex of household head here are problematic and I suggest to delete it.

OUR RESPONSE: We sincerely thank the reviewer for this comment. We are happy to read that Dr. Jiang has noted the substantial improvement in the manuscript following his and reviewer 2's very helpful comments in the first round. However, we are taken aback by Dr. Jiang's comment here. In most analyses that use data from household expenditure surveys, characteristics of the head of the household are often utilized as control variables. That is, they are used as additional covariates that account for (i.e. control for) other factors that might be driving the results. In other words, they control for confounding factors. Doing this allows us to isolate the effect on the outcome variable of the lone variable that we are interested in (in our case the variable of interest is price). This is the standard approach in the price elasticities of demand literature. This is also the approach that one of the authors (Dr. Grieve Chelwa) utilized in a recent study that is forthcoming in this journal. Characteristics of the head of the household are used as control variables precisely because household heads tend to exercise a lot of influence in the expenditure decisions of the household (there is a long and established literature on this). It is precisely because "Alcohol consumption and spending on different gender are quite different" (reviewer's quotes) that we need to control, as we do here, for their independent effects on consumption. Some households are headed by men and some by women and any differences emanating from this fact, and the reviewer suggests there might be some, ought to be accounted for. This is exactly what we do in our approach. Deleting them, as the reviewer suggests, would result in price elasticity estimates that are terribly contaminated by confounding factors.

It is also worth pointing out that the gender of the household head in no way assigns a gender to the household. (Assigning a gender to the household would certainly be meaningless). The gender of the household head merely tells us that the household head is either male or female, information that has a bearing on household decision making (as established in the literature and therefore requiring the need to account for it in the analysis).

In any case, we have decided to add the following sentence in the Method Section of the paper to add a little more clarity on why characteristics of the household head are useful in these kinds of analyses: "We use characteristics of the household head as demographic control variables because household heads are known to influence the household's expenditure decisions."

VERSION 2 – REVIEW

REVIEWER	Heng Jiang La Trobe University, Australia
REVIEW RETURNED	14-Feb-2019

GENERAL COMMENTS	I am happy with authors' response and all my comments have been addressed.
--